# Suppression of Peritoneal Fibrosis by Sonoporation of Hepatocyte Growth Factor Gene-Encoding Plasmid DNA in Mice

**DOI:** 10.3390/pharmaceutics13010115

**Published:** 2021-01-18

**Authors:** Koyo Nishimura, Koki Ogawa, Maho Kawaguchi, Shintaro Fumoto, Hidefumi Mukai, Shigeru Kawakami

**Affiliations:** 1Department of Pharmaceutical Informatics, Graduate School of Biomedical Sciences, Nagasaki University, 1-7-1 Sakamoto, Nagasaki-shi, Nagasaki 852-8588, Japan; koynishi0115@gmail.com (K.N.); k.ogawa.nagasaki.u@gmail.com (K.O.); bb55319402@ms.nagasaki-u.ac.jp (M.K.); hmukai@nagasaki-u.ac.jp (H.M.); 2Department of Pharmaceutics, Graduate School of Biomedical Sciences, Nagasaki University, 1-7-1 Sakamoto, Nagasaki-shi, Nagasaki 852-8588, Japan; sfumoto@nagasaki-u.ac.jp; 3Laboratory for Molecular Delivery and Imaging Technology, RIKEN Center for Biosystems Dynamics Research, 6-7-3 Minatojima-Minamimachi, Chuo-ku, Kobe, Hyogo 650-0047, Japan

**Keywords:** nanobubbles, ultrasound, gene therapy, peritoneal fibrosis, multicolor deep imaging, hepatocyte growth factor

## Abstract

Gene therapy is expected to be used for the treatment of peritoneal fibrosis, which is a serious problem associated with long-term peritoneal dialysis. Hepatocyte growth factor (HGF) is a well-known anti-fibrotic gene. We developed an ultrasound and nanobubble-mediated (sonoporation) gene transfection system, which selectively targets peritoneal tissues. Thus, we attempted to treat peritoneal fibrosis by sonoporation-based human HGF (hHGF) gene transfection in mice. To prepare a model of peritoneal fibrosis, mice were intraperitoneally injected with chlorhexidine digluconate. We evaluated the preventive and curative effects of sonoporation-based hHGF transfection by analyzing the following factors: hydroxyproline level, peritoneum thickness, and the peritoneal equilibration test. The transgene expression characteristics of sonoporation were also evaluated using multicolor deep imaging. In early-stage fibrosis in mice, transgene expression by sonoporation was observed in the submesothelial layer. Sonoporation-based hHGF transfection showed not only a preventive effect but also a curative effect for early-stage peritoneal fibrosis. Sonoporation-based hHGF transfection may be suitable for the treatment of peritoneal fibrosis regarding the transfection characteristics of transgene expression in the peritoneum under fibrosis.

## 1. Introduction

Chronic kidney disease (CKD) patients are increasing worldwide, with over 10% of the world population affected [1], and end-stage CKD patients are expected to increase. Peritoneal dialysis is a replacement therapy for patients with end-stage CKD. Peritoneal dialysis has some advantages over hemodialysis, such as maintenance of kidney function, lifestyle flexibility, and independence. Contrarily, long-term peritoneal dialysis often causes peritoneal fibrosis, which leads to the interruption of treatment. Although glucocorticoids and tamoxifen have been shown to have beneficial effects [2], curative therapy for peritoneal fibrosis has not been developed. Thus, to improve the quality of life of peritoneal dialysis patients, the development of prevention and therapeutic strategies for peritoneal fibrosis is essential.

Gene therapy is a potent approach for peritoneal fibrosis therapy because it can target a wide range of molecules and is expected to act for a longer time than small molecule or protein drugs. Transfection methods for gene therapy can be divided into viral or non-viral vectors. Within a non-viral vector, plasmid DNA (pDNA) is promising in terms of safety and efficiency, but the transfection ability of pDNA itself is very low. Thus, a transfection system is required for gene therapy using pDNA. Electrostatic-based pDNA complexes with cationic liposome/polymers are popular carriers for pDNA-based gene transfection. Contrarily, external stimuli (e.g., light [3], mechanical stimuli [4,5,6], and magnetic field [7]) have been studied as driving forces of site-specific gene transfection. Sonoporation is a promising gene transfection system based on ultrasound stimuli [8,9,10]. In sonoporation, cavitation energy is generated by ultrasound irradiation of bubble formulation. During this process, transient pores are created on the plasma membrane, and genes are transfected into the cytosol [11,12]. We have developed a transfection system for peritoneal tissues, which is based on an intraperitoneal injection and sonoporation [13,14]. Moreover, this method achieved efficient and safe transfection selectively for the mesothelial cells on the intraperitoneal tissue surface of normal mice without transfection in the submesothelial layer.

Epithelial-mesenchymal transition (EMT) of mesothelial cells is a key process in the progression of peritoneal fibrosis [15]. Suppressing EMT by an intraperitoneal sonoporation system in mesothelial cells is expected to be an effective prevention method for peritoneal fibrosis. During EMT, mesothelial cells invade the submesothelial layer, and mesothelial cells and fibroblasts in the submesothelial layer are transformed into myofibroblasts and produce collagen fibers. Considering the histological changes induced by EMT, it is expected that the target cell can be different before and after EMT. Before EMT, which is used for preventive gene therapy, the mesothelial surface layer can be a target. Because the sonoporation-based transfection system we developed could selectively transfect the mesothelial surface layer, preventive therapy is expected to be suitable.

After EMT, which is used for curative gene therapy, it is necessary to transfect the submesothelial layer, which lies in the mesothelial cells and fibroblasts. However, the optimized sonoporation condition cannot transfect genes through the junctions between mesothelial cells of the intact peritoneum [13]. At the beginning of EMT, the cell–cell junctions (adhesion junctions, tight junctions) and basal membranes are disrupted [16]. Therefore, we hypothesized that sonoporation might enable the transfer of genes to the submesothelial layer during the early stage of peritoneal fibrosis. Therefore, a curative approach for peritoneal fibrosis is expected from gene therapy.

Some anti-fibrosis proteins that inhibit EMT and ameliorate peritoneal fibrosis have been proposed to date. Hepatocyte growth factor (HGF) is an anti-fibrotic protein that acts via inhibition of TGF-β signaling. HGF exposure to peritoneal mesothelial cells prevented fibrosis induced by high glucose [17]. HGF secreted by injected mesenchymal stem cells (MSCs) ameliorated chlorhexidine digluconate (CG)-induced peritoneal fibrosis [18]. Thus, we estimated that sonoporation-based HGF gene transfection exhibited preventive and curative effects.

In this study, we chose human HGF (hHGF) as a therapeutic gene. First, we verified whether peritoneal fibrosis can be prevented by single hHGF gene transfection using a sonoporation-based method in normal mice with subsequent CG challenge. Next, we evaluated the transfection efficacy and the intra-tissue distribution of transgene expression in the peritoneum causing peritoneal fibrosis at an early stage by sonoporation-based transfection of the reporter gene, luciferase, or ZsGreen1. Finally, we verified the curative effects of peritoneal fibrosis in the early stage by hHGF transfection using sonoporation.

## 2. Materials and Methods

### 2.1. Animals and Peritoneal Fibrosis Models

Male ddY mice (five-week-old, 26–33 g) were purchased from Japan SLC, Inc. (Hamamatsu, Japan). All animal experiments were performed in accordance with the guidelines for animal experimentation of the Nagasaki University (approval number: 1308051086–6 (2016)).

To prepare the peritoneal fibrosis model mice, 0.1% *w/v* chlorhexidine gluconate (CG) (Sigma–Aldrich) in 15% ethanol and 150 mM sodium chloride was 0.1 mL/g intraperitoneally injected [19]. To prepare the early-stage peritoneal fibrosis model, CG was injected into the mice for three consecutive days.

### 2.2. pDNA

pCMV-Luc was constructed as previously described [20]. The hHGF expression vector without any CpG, pCpGfree-hHGF was custom-made by subcloning the CpG-depleted hHGF gene into the pCpGfree-mcs vector (InvivoGen, San Diego, CA, USA). pCMV-Luc and pZsGreen1-N1 (Takara Bio Inc., Shiga, Japan) were amplified in the *Escherichia coli* strain DH5α. pCpGfree-LacZ (InvivoGen, San Diego, CA, USA) and pCpGfree-hHGF were amplified in the *E. coli* strain GT115. Endofree Plasmid Giga Kit (QIAGEN GmbH, Hilden, Germany) was used to purified pDNA.

### 2.3. Preparation of Nanobubbles

Nanobubbles were prepared according to a previous study [13,21]. Briefly, DSPC (Avanti Polar Lipids) and mPEG_2000_-DSPE (NOF) dissolved in methanol (DSPC: mPEG_2000_-DSPE = 94:6 (m/m)) were dried by evaporation and vacuum desiccated. The lipid film was dispersed in phosphate-buffered saline (PBS) at 65 °C, then sonicated for 3 min using an ultrasonic homogenizer (US 300E, Nissei, Tokyo, Japan). The liposome was enclosed with a 7.5 mL perfluoropropane gas and shaken in a sonicator (BRANSONIC B1200, Emerson Electric Co., St. Louis, MO, USA).

### 2.4. In Vivo Intraperitoneal Gene Transfection

Before transfection, anesthesia was induced by intramuscular injection of three mixed anesthetic agents [22]. For transfection by sonoporation, a mixture of 60 µg of pDNA and 250 µg of nanobubbles, dissolved in 600 μL of PBS were injected intraperitoneally into the mice. Immediately, the abdominal area was irradiated transdermally with ultrasound (frequency: 1.045 MHz; duty cycle: 50%; burst rate: 10 Hz; intensity: 1.0 W/cm^2^) using a sonicator (Sonopore-4000, Nepa Gene, Chiba, Japan) with a probe (diameter, 20 mm). For transfection with pDNA/PEI complex (cationic polyplex), pDNA (60 µg), and in vivo-jetPEI (PolyPlus-Transfection, Illkirch, France) were mixed at an N/P ratio of 8. The complex (total volume: 1 mL) was injected intraperitoneally into mice.

### 2.5. Quantification of Transgene Expression

Luciferase expression was measured 6 h after transfection, as described previously [23]. For the measurement of hHGF, the peritoneal wall and peritoneal fluid were harvested from mice. The peritoneal wall was homogenized by three times the weight of RIPA buffer (50 mM Tris-HCl, pH 7.4; 75 mM NaCl; 0.5% Triton-X; 0.25% sodium deoxycholate; 0.05% SDS). hHGF was measured using the Human HGF Quantikine ELISA Kit (R&D Systems). Total protein (450 μg) for the peritoneal wall and 50 μL of the peritoneal wall were used for ELISA. The measurements were conducted according to the manufacturer’s instructions.

### 2.6. Peritoneal Equilibration Test

Peritoneal dialysis fluid (Dianeal PD-4 4.25 peritoneal dialysis fluid (PDF) (Baxter, Tokyo, Japan) was intraperitoneally injected into mice. Peritoneal fluid (D0) was collected immediately from the mice. The peritoneal fluid (D2) and plasma (P2) were collected 2 h after injection of PDF. Glucose and creatinine concentrations of samples were measured using LabAssay^TM^ Glucose (Wako) and LabAssay^TM^ Creatinine (Wako), respectively. The ratios of D2/D0 and D2/P2 were calculated.

### 2.7. Masson’s Trichrome Staining

The peritoneal walls were collected and fixed with 4% paraformaldehyde in PBS. They were then embedded in O.C.T. compound and frozen at 80 °C. They were placed into 5 µm sections using a microtome (CM1950; Leica Microsystems GmbH, Wetzlar, Germany) and stained with Masson’s trichrome stain. The stained samples were observed using Axio Vert.A1 microscope (Carl Zeiss, Oberkochen, Germany) with a 20× objective lens. The thickness of the submesothelial layer was calculated using ImageJ software.

### 2.8. Multicolor-Deep-Imaging Analysis

Multicolor deep imaging was performed based on tissue clearing, as described previously [14]. To observe the invasion of mesothelial cells, 300 µL of 100 µM 1,1-Dioctadecyl-3,3,3′,3-tetramethylindocarbocyanine perchlorate (DiI, Sigma–Aldrich) was intraperitoneally injected into mice 1 d before preparing the peritoneal fibrosis model. Then, 300 µL of 100 µM 1,1′-Dioctadecyl-3,3,3′,3′-tetramethylindodicarbocyanine (DiD, Invitrogen) was intraperitoneally injected into mice 1 h before tissue clearing. To observe the depth of transgene expression, pZsGreen1-N1 was transfected by sonoporation 1 d after three consecutive days of CG injections. Then, 1 h before tissue clearing, 300 µL of 100 µM DiI was intraperitoneally injected into the mice. Collected samples were cleared using the Sca*l*eSQ(0) method [24].

### 2.9. Hydroxyproline Assay

Twelve molars hydrochloric acid was added to a 10 mg sample of the homogenized peritoneal wall and hydrolyzed at 120 °C for 3 h. The hydrolyzed samples were centrifuged at 10,000× *g* for 5 min. The hydroxyproline concentration of the supernatant was measured using a hydroxyproline assay kit (Merck KGaA, Darmstadt, Germany).

### 2.10. Statistical Analysis

ANOVA was used to analyze the statistical significance of the differences among the groups. Tukey’s test was used for multiple comparisons between all the groups. Dunnett’s test was used for multiple comparisons between the control and treatment groups. A difference with *p* < 0.05 was considered significant.

## 3. Results

### 3.1. Transgene Expression Periods of hHGF in Peritoneal Fibrosis Mice

To evaluate the duration of hHGF gene expression in the peritoneal cavity, we measured hHGF protein levels in the peritoneal wall and peritoneal fluid by ELISA. One day after transfection, the hHGF protein level was significantly higher than that on day 0. After that, hHGF expression decreased to the same level as that on day 0 (Figure 1).

### 3.2. Prevention Effects of Sonoporation-Mediated hHGF Transfection on the Progression of CG-Induced Peritoneal Fibrosis

To evaluate the preventive effect of hHGF transfection by sonoporation on peritoneal fibrosis, we transfected the hHGF gene by sonoporation followed by 14 days of convective CG injection (Figure 2a). One day after the last CG injection, we measured the hydroxyproline level in the peritoneal wall as collagen synthesis. As a result, hydroxyproline concentration in hHGF-transfected mice by sonoporation tended to be lower than that of only CG-injected mice, although a significant difference was not detected (*p* = 0.096, analyzed by Tukey’s multiple comparison test) (Figure 2b).

Next, Masson’s trichrome staining of the peritoneal wall was performed to evaluate peritoneal thickness. In the CG only group, the control vector (LacZ) + ultrasound and hHGF—ultrasound increased the thickness of the submesothelial layer (Figure 2c,d). In contrast, in the hHGF + ultrasound group, thickening of the submesothelial layer was significantly suppressed compared to that in the other CG-injected group at day 14 (Figure 2c,d).

The peritoneal function was evaluated using a peritoneal equilibration test. After day 14 of CG injections, the D2/D0 glucose ratio and D2/P2 creatinine ratio were measured for glucose absorption ability and creatinine elimination rate. The D2/D0 glucose ratio of CG-injected mice was lower than that of control mice. hHGF transfection by sonoporation partially improved the D2/D0 glucose ratio, although it did not reach the level observed in control mice (Figure 2e). Contrarily, the D2/P2 creatinine ratio of the hHGF + ultrasound group was lower than that of CG only group or LacZ + ultrasound group (Figure 2f).

### 3.3. Evaluation of Early-Stage Peritoneal Fibrosis

After three consecutive days of CG injections, the spatial distribution of mesothelial cells was evaluated by labeling mesothelial cells and peritoneal surface with DiI and DiD, respectively, followed by the multicolor-deep-imaging analysis. In normal mice, DiI-labeled mesothelial cells were located at the DiD-labeled surface layer, whereas in CG-injected mice, some mesothelial cells were located under the mesothelial layer. (Figure 3).

### 3.4. Transfection Efficiency in Early-Stage Peritoneal Fibrosis

We evaluated the influence of CG injection duration on transfection efficiency in the peritoneal tissues. Luciferase, as a reporter gene, was transfected to the following mice: CG(-) mice, 3 days CG-injected mice, and 7 days CG-injected mice using sonoporation or in vivo jet-PEI (Figure 4a). In the case of sonoporation-mediated transfection, transfection efficiency did not change between CG(-) mice and 3 days CG-injected mice at any peritoneal organ (Figure 4b). In 7 days CG-injected mice, transfection efficacy in the peritoneal wall was significantly decreased (Figure 4b). In contrast, when mice were transfected with in vivo jet-PEI, luciferase expression in CG-injected mice tended to be lower than that of intact mice in the liver (Figure 4c). In other peritoneal organs, luciferase expression was below the limit of quantitation (Figure 4c).

### 3.5. Distribution of Transgene Expression in the Early Stage of Peritoneal Fibrosis

We evaluated the distribution of transgene expression in peritoneal tissues using a multicolor-deep-imaging method after sonoporation-mediated transfection of ZsGreen1 in early-stage peritoneal fibrosis mice. After three consecutive days of CG injections was followed by gene transfection by sonoporation, the normal site and injury site were observed. At the normal site, transgene expression was observed in the tissue surface cells (Figure 5a). Conversely, in the injury site, transgene expression was not observed on the tissue surface but in the inner tissue (Figure 5b). In the case of 7 days injection of CG, the transgene expression was almost not observed (Figure 5c). When mice were transfected using in vivo jet-PEI after 3 days CG injection, transgene expression was not observed (Figure 5d).

### 3.6. Suppression Effects of Sonoporation-Mediated hHGF Transfer on the Progression of Early-Stage Peritoneal Fibrosis

To evaluate the effect of early treatment on the progression of peritoneal fibrosis, hHGF was transfected after three consecutive days of CG injections, and Masson’s trichrome staining and peritoneal equilibration test were conducted on the day following the 14-consecutive days of CG injections (Figure 6a). The result was similar to that of the preventive effects (Figure 2). The increase in the thickness of the peritoneum was significantly suppressed by hHGF transfection (Figure 6b,c). Moreover, both the D2/D0 glucose ratio and D2/P2 creatinine ratio were significantly improved by hHGF transfection (Figure 6d,e).

## 4. Discussion

Carriers for intraperitoneal pDNA or siRNA delivery have been developed, such as cationic liposomes and cationic gelatin in mice [25,26]. These methods transfer pDNA to the mesothelial cells of the intact peritoneum via electrostatic interactions, and they showed preventive effects on CG-induced peritoneal fibrosis. However, the histological characteristics of peritoneal fibrosis are different from those of the intact peritoneum. Thus, we consider that the optimal transfection system and therapeutic gene should be selected depending on the state of the peritoneum. In this study, we demonstrated a preventive effect against CG-induced peritoneal fibrosis. Besides that, we tried for a curative approach based on the characteristics of sonoporation-mediated intraperitoneal transfection, which this study revealed. As a therapeutic gene, we chose hHGF because it is known to suppress tissue fibrosis, including peritoneal fibrosis, by attenuating TGF-β signaling [27,28,29].

First, we evaluated hHGF expression in the peritoneal wall and peritoneal fluid during the preventive challenge. One day after transfection, hHGF levels in the peritoneal wall and peritoneal fluid were higher than those on day 0 (Figure 1). After that, hHGF expression decreased to a level comparable to that on day 0. Other groups and we have reported that the pCpGfree vector can sustain the gene expression term over 28 days by avoiding the immune response of pDNA [13,30]. In all of these reports of the pCpGfree vector, transgene expression was evaluated in normal mice. However, there are no reports on the absence of CpG on immune-activated states, such as fibrosis. In this study, CG was injected after pCpGfree-hHGF was transferred by sonoporation. CG injection has been reported to promote TGF-β1 production, resulting in the activation of immune reactions. Thus, we speculated that hHGF expression was transient due to immune activation by CG injection. Contrarily, the pCpGfree vector can transfer genes with a low immune response; therefore, we believe that using the pCpGfree vector is advantageous for the therapy of peritoneal fibrosis.

We evaluated the preventive effects of sonoporation-mediated hHGF transfection. Fourteen days after hHGF transfection, hydroxyproline level in the peritoneum, the thickness of the submesothelial layer, and peritoneal function measured by the peritoneal equilibration test were evaluated as therapeutic criteria. Hydroxyproline levels tended to be suppressed by sonoporation-based hHGF transfection (Figure 2b). Additionally, Masson’s trichrome staining of the peritoneum showed that the thickness of the peritoneum was also suppressed (Figure 2c,d). About the peritoneal equilibration test, the D2/D0 glucose ratio and the D2/P2 creatinine ratio indicate the removal efficacy of water and the transportation of small solutes, respectively [31]. Both indexes were improved by the hHGF + ultrasound group compared to the CG-injected control group (Figure 2e,f). These results correspond with the report by Ueno et al. [18], in which HGF-expressing MSCs exerted a preventive effect on the peritoneal fibrosis model. Thus, sonoporation-based hHGF transfection may prevent CG-induced peritoneal fibrosis.

Lua et al. reported that three consecutive days of CG injections induced the loss of cell–cell junction interactions and mesothelial cell infiltration [32]. To confirm the pathology of early-stage peritoneal fibrosis, we attempted to observe the invasion of mesothelial cells into the submesothelial layer. We have developed a multicolor-deep-imaging analysis method [33] that co-visualizes objects with biological structures (e.g., blood vessels [4,34], ventricle walls [35], and mesothelial cells [13,14]) by combining tissue clearing and confocal laser scanning microscopy (CLMS) Z-stack imaging. In a previous report, we labeled mesothelial cells by intraperitoneal DiI injection. In this study, we observed the distribution of DiI-labeled mesothelial cells in CG mice. Before CG injection, mesothelial cells were stained with intraperitoneal DiI injection in intact mice [13,14]. Following CG injection, DiD was injected intraperitoneally to stain the peritoneal surface before tissue excision. In CG-injected mice, DiI-labeled mesothelial cells were located below the DiD-labeled peritoneal surface, suggesting that mesothelial cells started to invade the submesothelial layer after three consecutive days CG injection (Figure 3). These results are consistent with a previous report in which EMT was analyzed using transgenic mice [32]. The multicolor-deep-imaging analysis data support the view that CG injection for three consecutive days can be an early stage of the peritoneal fibrosis model in mice.

In the early stage of peritoneal fibrosis, the junction of mesothelial cells is loosened. To date, there is no information on the characteristics of transgene expression in peritoneal fibrosis. We evaluated the transfection efficiency in the peritoneal organs after CG injection. Transfection efficiency by sonoporation was not changed by prior 3 days CG exposure, although prior 7 days CG exposure decreased the transgene expression level of peritoneal walls (Figure 4b). Transfection of mice with prior 7 days CG exposure may be difficult because it is expected that mesothelial cells shed into the submesothelial layer, and fibroblasts produce collagen through EMT [19]. In contrast, the transfection efficiency of in vivo-jetPEI was drastically decreased in mice with prior CG exposure (Figure 4c). In our previous study, we revealed that the present sonoporation-based transfection system delivered genes only to the peritoneum that covers the liver and stomach, not to the parenchyma of organs in normal mice [13]. Considering that, the transgene expression in liver, stomach, and spleen in Figure 4a was probably derived from the peritoneum covering the organs, not from the parenchyma, even in the mice with peritoneal fibrosis. It is thought that the peritoneum on peritoneal organs functions as an additive source to provide HGF to the intraperitoneal cavity. These results suggest that sonoporation is a suitable transfection method for the peritoneum with early-stage fibrosis because transgene expression by sonoporation may be unaffected by peritoneal fibrosis.

Therefore, we evaluated the distribution of transgene expression obtained by sonoporation-based transfection from the viewpoint of the depth of transfected cells in the peritoneal tissues. ZsGreen1 expression was observed on the inner side of the peritoneum when ZsGreen1 was transferred by sonoporation after the 3 days of CG injection (Figure 5b). The distribution of transgene expression is different from our previous result, in which the distribution of transgenes was restricted to the peritoneal surface of intact mice [13]. We also observed the distribution when the gene was transferred after 7 days of CG injection; however, only a few ZsGreen1 were detected (Figure 5c). As a conventional transfection reagent, we used a commercially available gene transfer reagent, in vivo-jetPEI, to compare the distribution of transgene expression. In Figure 5d, the transgene expression of ZsGreen1 was hardly observed when the gene was transferred by in vivo-jetPEI after 3 days of CG injection. Transgene expression observed by multicolor-deep-imaging corresponded with the luciferase expression analysis (Figure 4). Thus, multicolor-deep-imaging analysis suggests that sonoporation transfects to the submesothelial layer in early-stage peritoneal fibrosis, although it has low transfection capacity for advanced fibrosis. Thus, we considered that sonoporation-based gene delivery can be applied as a curative therapy for early-stage peritoneal fibrosis, especially in mice exposed to 3 days CG before transfection.

Considering the transfection characteristics of early-stage peritoneal fibrosis (Figure 4 and Figure 5), we evaluated the suppressive effect of sonoporation-based hHGF transfection on 3 days CG-injected mice. We performed Masson’s trichrome staining and peritoneal equilibration tests to evaluate the thickness of the peritoneum and peritoneal function. Consequently, CG-induced thickening of the submesothelial layer was significantly suppressed by hHGF transfection (Figure 6b,c). Additionally, both the D2/D0 glucose ratio and D2/P2 creatinine ratio were significantly improved compared to the CG-injected group with the control treatment (Figure 6d,e). These results suggest that sonoporation-based hHGF transfection could also be applied for curative therapy to suppress early-stage peritoneal fibrosis besides preventive application. Recently, early diagnostic methods for peritoneal fibrosis have been studied [36]. To date, there have been reports on the prevention of peritoneal fibrosis by advanced hHGF transfection [17], but there have been no reports on the curative approach for peritoneal fibrosis progression in the early stage of peritoneal fibrosis. Thus, we succeeded in obtaining the treatment effects of hHGF transfection in the early stages of peritoneal fibrosis.

In this study, we provided a preclinical proof of concept for preventing and treating peritoneal fibrosis by hHGF transfection using sonoporation. In hospitals, peritoneal dialysis patients are regularly injected with dialysis fluid into their peritoneal cavity. Thus, we expect a clinical protocol in which bubbles and hHGF-coding vector are delivered to patients, especially receiving peritoneal dialysis, by dissolving in dialysis fluid, followed by the ultrasound irradiation to the abdomen. In addition, to verify transgene expression clinically, the nuclear medicine approach can be an alternative to fluorescence imaging, because it allows visualizing the distribution of transgene expression quantitatively in a minimally invasive manner using Na/I symporter reporter gene and radioactive ^124^I [37].

## 5. Conclusions

Gene transfection into the fibrotic peritoneum is difficult because mesothelial cells shed into the submesothelial layer through EMT. It is evident from our result that the transfection efficiency of in vivo-jetPEI was very low in CG-injected mice. In contrast, we have shown that sonoporation maintains transfection efficiency in early-stage peritoneal fibrosis. Further, by using multicolor deep imaging, we clarified that the transgene is expressed in the submesothelial layer after the gene is transferred to early-stage fibrosis. Based on the information on transfection characteristics, we succeeded in suppressing early-stage fibrosis by sonoporation-mediated hHGF transfection; that is, sonoporation can be applied not only for prevention but also for treatment. We believe that this therapeutic system is potent in suppressing the aggravation of peritoneal fibrosis caused by peritoneal dialysis.

## Figures and Tables

**Figure 1 pharmaceutics-13-00115-f001:**
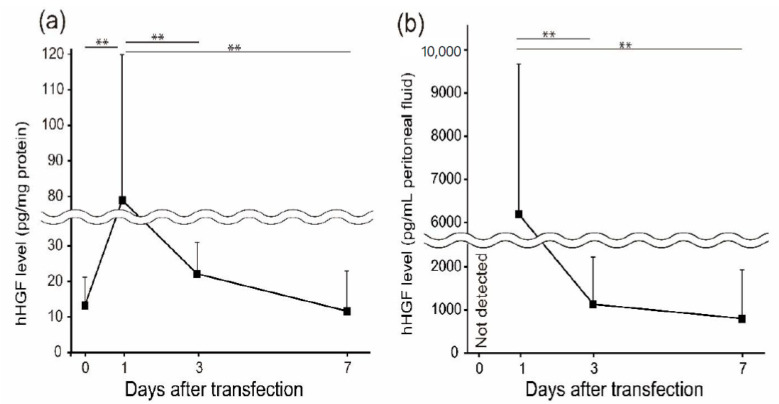
Human hepatocyte growth factor (hHGF) expression periods of chlorhexidine digluconate (CG)-induced peritoneal fibrosis by sonoporation-mediated hHGF transfection. Following hHGF transfection, CG was injected every day. The hHGF expression level in the peritoneal wall (**a**) and peritoneal fluid (**b**) was analyzed on days 1, 3, and 7 after transfection. Values are expressed as the mean + SD (*n* = 5). ** *p* < 0.01 by Tukey’s multiple comparison test.

**Figure 2 pharmaceutics-13-00115-f002:**
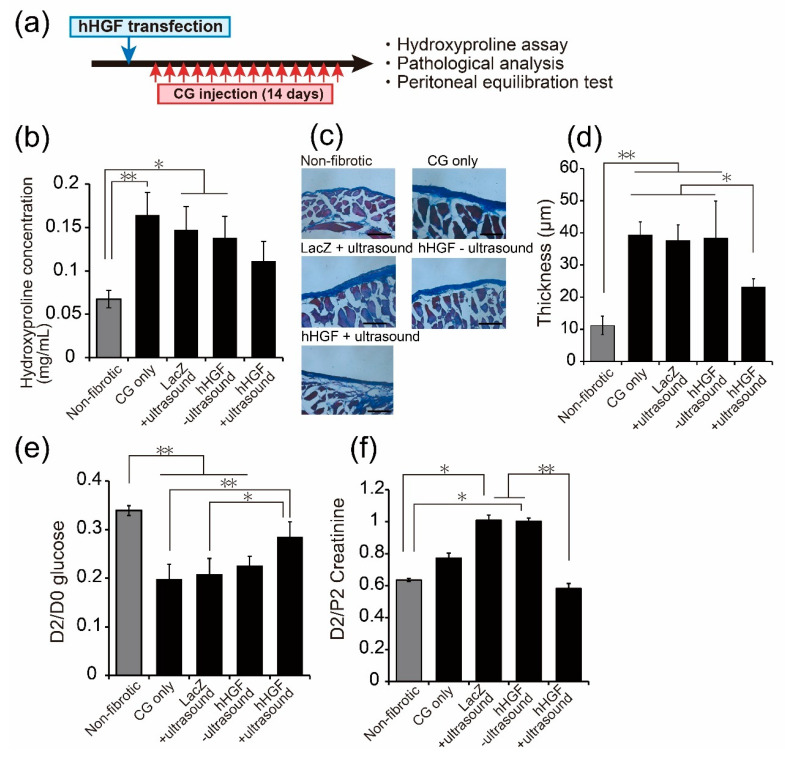
Prevention effects of the progression of CG-induced peritoneal fibrosis by sonoporation-mediated hHGF transfection. (**a**) Experiment schedule. Mice were transfected hHGF and injected CG for 14 days from the following day of transfection. (**b**) The hydroxyproline concentration in the peritoneal wall (**c**) Masson’s trichrome staining of the peritoneal wall. Scale bar represents 100 µm. (**d**) The thickness of the submesothelial layer. Data are represented as mean ± SD (*n* = 3). (**e**,**f**) The peritoneal absorption of glucose and creatinine from the dialysate (peritoneal fluid 2 h (D2)/D0) glucose and the dialysate-to-plasma ratio (D2/plasma 2 h (P2)) creatinine were examined by peritoneal equilibration test. Values are expressed as the mean ± SD (*n* = 3–4). * *p* < 0.05, ** *p* < 0.01 by Tukey’s multiple comparison test.

**Figure 3 pharmaceutics-13-00115-f003:**
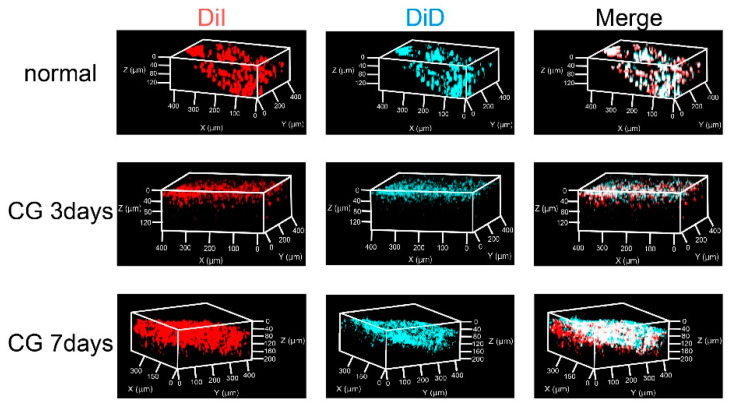
Evaluation of the migration of mesothelial cells in early-stage peritoneal fibrosis. Spatial distribution of mesothelial cells in the peritoneum after CG injection. DiI solution was intraperitoneally injected before CG treatment to label mesothelial cells that were originally located on the surface of the peritoneal wall. DiD solution was intraperitoneally injected before tissue clearing to label the surface of the peritoneum. Each specimen was observed with confocal laser scanning microscopy (CLMS) following to cleared with Sca*l*eSQ(0) method.

**Figure 4 pharmaceutics-13-00115-f004:**
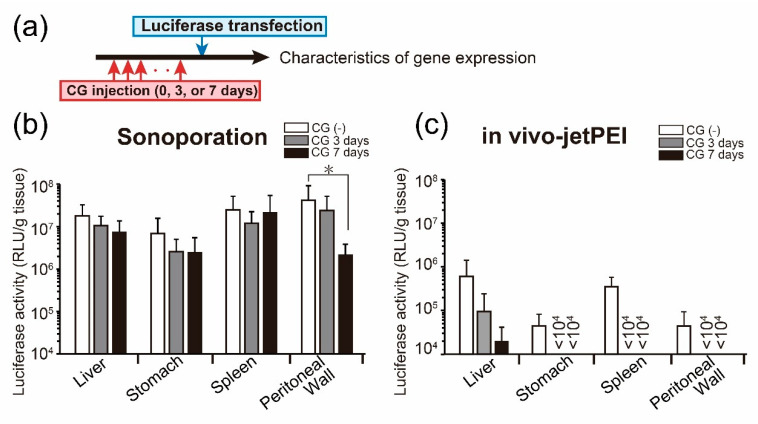
Evaluation of transgene expression efficacy in the early stage of peritoneal fibrosis. Luciferase gene was transferred to CG(-) mice, 3 days CG-injected mice, or 7 days CG-injected mice using sonoporation (**b**) or in vivo jet-PEI (**c**). (**a**) Experiment schedule. Luciferase expression was evaluated 6 h after transfection. Values are expressed as the mean + SD (*n* = 3–8). * *p* < 0.05 by Dunnett’s multiple comparison test for each organ (vs. CG(-)).

**Figure 5 pharmaceutics-13-00115-f005:**
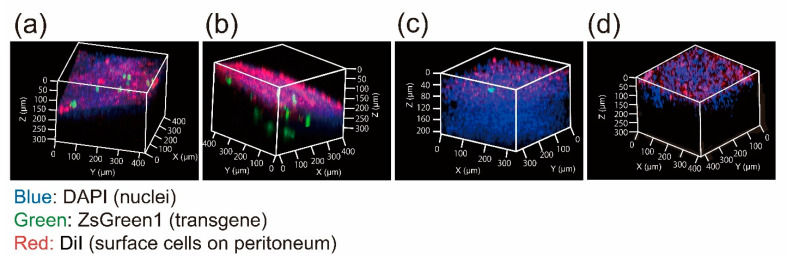
The multicolor deep imaging of transgene expression in the early-stage peritoneal fibrosis. ZsGreen1 was transfected to the mice injected CG for 3 days by sonoporation. The transgene expression was observed in the normal site (**a**) and injure site (**b**), respectively. (**c**) Transgene expression after sonoporation to the mice CG-treated for 7 days. (**d**) The mice CG-treated for 3 days were transfected by ZsGreen1 by in vivo-jetPEI.

**Figure 6 pharmaceutics-13-00115-f006:**
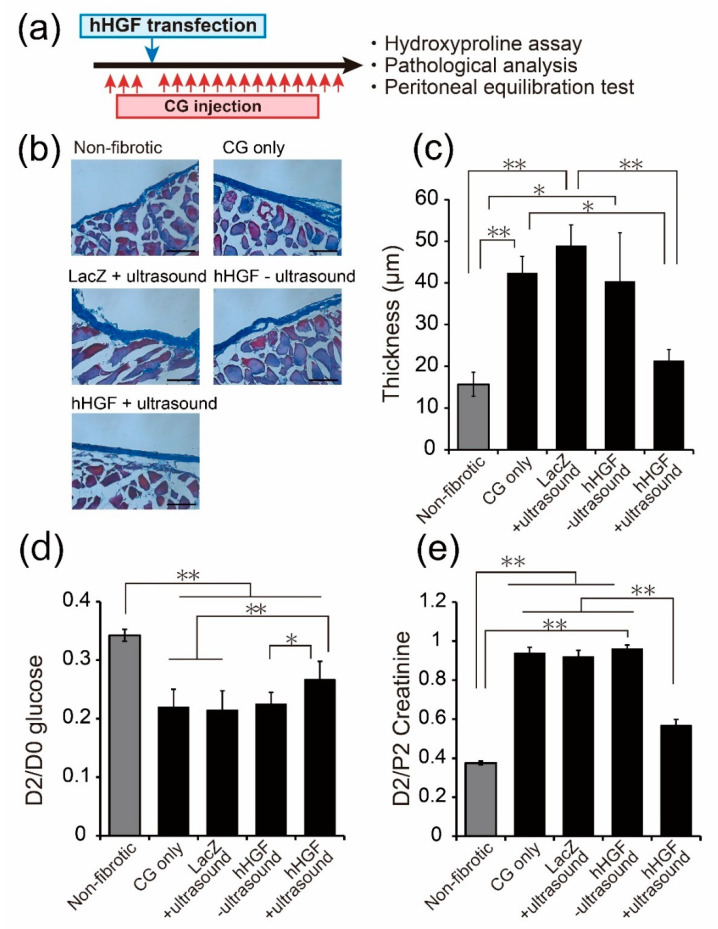
Suppression effects of early-stage peritoneal fibrosis by sonoporation-mediated hHGF transfection. (**a**) Experiment schedule. hHGF was transfected to the 3 days CG-injected mice by sonoporation, followed by the 14 days CG injection. (**b**) Masson’s trichrome staining of the peritoneal wall. Scale bar represents 100 µm. (**c**) The thickness of the submesothelial layer. Data are represented as mean ± SD (*n* = 3). (**d**,**e**) The peritoneal absorption of glucose and creatinine from the dialysate (D2/D0) glucose and the dialysate-to-plasma ratio (D2/P2) creatinine were examined by peritoneal equilibration test. Values are expressed as mean ± SD (*n* = 3–4). * *p* < 0.05, ** *p* < 0.01 by Tukey’s multiple comparison test.

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
