# Peer review of "Suppression of Peritoneal Fibrosis by Sonoporation of Hepatocyte Growth Factor Gene-Encoding Plasmid DNA in Mice"

_pharmaceutics, 2021, doi:10.3390/pharmaceutics13010115_

Round 1
Reviewer 1 Report
The manuscript of Nishimura et al described a novel ultrasound and nanobubble method (sonoporation) for gene delivery to peritoneal tissues. They used human hepatocyte growth factor (HGF) as a therapeutic gene to evaluate its preventive and curative effects on suppression peritoneal fibrosis in a mice model induced by intraperitoneal infection of chlorhexidine gluconate. This study implies a gene therapy approach to suppress of suppression of peritoneal fibrosis by means of the novel sonoporation. Reviewer recommends the publication of this manuscript.
Main concern: Figure 4 suggests the sonoporation transfection deliveres reporter gene expression to liver, stomach and spleen as efficiently to peritoneal wall. Will the transient expression of HGF in the non-targeted tissues potentially impacts the application of preventive and curative approach via intraperitoneal injection of HGF gene?
Minors:
- Line2 115-123: 4 In vivo intraperitoneal gene transfection. The mothed was not described clearly: it lacks the volume of the transgene complex used for mice injection;
- Line 137: “PD” should be PDF;
- Line 145: “Axio Vert. A1 microscope” should be “Axio Vert.A1 microscope”;
- Line 187: “LacZ +US”, the “US” should be specified and also in the legend of figure 2. Reviewer assumes it is for “ultrasound”?
- Line 122 “in vivo jetPEI” and line 213 and others “ in vivo jet PEI” should be “ in vivo-jetPEI”; Also correct those in figure 4 and its legend;
- Line 263-264 (Figure 4): the panel "a" is wrong, it is not hGHF transfection but reporter gene, and “stomack” should be “stomach”
Reviewer 2 Report
The manuscript « Suppression of peritoneal fibrosis by sonoporation of hepatocyte growth factor gene-encoding plasmid DNA in mice” by Nishimura et al. describes the utilization of sonoporation to deliver a therapeutic gene to the peritoneal cavity. Hepatocyte growth factor is delivered in a mouse model of peritoneal fibrosis and the paper describes both preventive and curative protocols. In addition, state of the art fluorescent imaging is presented to document and locate gene transfer.
This is in my view an excellent manuscript, well documented and convincing.
Minor modifications would in my view improve the manuscript:
Figure 1: The scale of the Y axis of the figure is such that it is difficult to evaluate the level of HGF expression at days 3 and 7. I would suggest to try to reduce the height of the Y axis. Presenting the SEM instead of SD would reduce the size of the error bars and should in my view provide a more informative figure.
Figure 2: The term control is a little bit misleading. I would suggest “non-fibrotic mice”.
Figure 4: spelling: stomach instead of stomack.
In the discussion, the authors should in my view discuss whether this technique is applicable to humans. I think that it is the case but that should be emphasized. A key step would be to verify whether transgene expression could be achieved in humans and fluorescence imaging is unlikely to be used in humans? Or is it? Maybe nuclear medicine methods (as in J Nuclar Medicine 2010 51:951) may be more applicable. A small paragraph discussion these points would in my view enrich the manuscript.
